# Explainability Guided COVID-19 Detection in CT Scans

**Ameen Ali**                                                    AMEENALI@MAIL.TAU.AC.IL
**Tal Shaharabany**[†]                                           SHAHARABANY@MAIL.TAU.AC.IL
**Lior Wolf**                                                    WOLF@MAIL.TAU.AC.IL
*Blavatnik School of Computer Science, Tel-Aviv University, Israel*

## Abstract

Radiological examination of chest CT is an effective method for screening COVID-19 cases. In this work, we overcome three challenges in the automation of this process: (i) the limited number of supervised positive cases, (ii) the lack of region-based supervision, and (iii) variability across acquisition sites. These challenges are met by incorporating a recent augmentation solution called SnapMix, a novel explainability-driven contrastive loss for patch embedding, and by performing test-time augmentation that masks out the most relevant patches in order to analyse the prediction stability. The three techniques are complementary and are all based on utilizing the heatmaps produced by the Class Activation Mapping (CAM) explainability method. State-of-the-art performance is obtained on three different datasets for COVID detection in CT scans.

## 1. Introduction

Deep neural networks are currently the leading image classification method. Their ability to generalize is well-documented. However, in many medical imaging domains, one faces challenges that reduce the effectiveness of generic solutions. First, due to the cost of acquisition, privacy issues, and the expertise required for labeling, typical datasets are smaller than those available for many other computer vision tasks. Second, in medical images, the exact capturing apparatus, its setting and its operators can all greatly affect the distribution of the obtained images, causing a sizable domain shift. Third, many diseases are manifested through symptoms that are well localized, while supervision is given at the image level.

In this work, we demonstrate that explainability methods, which link the classification outcome to specific image regions, can provide an important building block for overcoming these three issues. First, the heatmap obtained from such methods serves as the basis for an augmentation method called SnapMix (Huang et al., 2021), which we demonstrate to be also effective for the COVID-19 classification task we study in this work. Second, the heatmap can provide a delineation of whether or not local image patches are strongly linked to the obtained classification. By requiring image patches of similar relevancy to have similar embedding, we can improve the classification performance. Third, we can use the heatmap to validate, at test time, the stability of the obtained classification, by perturbing the image locations most relevant to the prediction. If the majority of perturbations do not support the prediction, we flip the predicted label.

We evaluate our method with well-established benchmarks for the classification of Computed Tomography (CT) scans as COVID-19 positive or COVID-19 negative, and present

---

[†] Part of the Ph.D of Tal Shaharabany

clear evidence of the utility of our method. The gap in performance we obtain is larger than the variance between state-of-the-art methods. On one site, in which performance (F1 score and accuracy) is over 90%, we improve to over 95%. On a second site, in which performance levels are around 80%, we obtain results of almost 90%. In a third dataset, for which performance almost saturates, we reduce the error rate by one and a half or more, depending on the measurement error.

## 2. Related Work

**COVID19 Classification** The SARS COV-2 infection (COVID-19) has a devastating impact on the respiratory system and has caused an enormous number of deaths. Over the last year, many deep learning methods were developed for classifying COVID-19 in 2D or 3D medical images (Gozes et al., 2020; Rahimzadeh and Attar, 2020; Zhang et al., 2020; Wang et al., 2020a). Some recent methods use transfer learning from models pretrained on ImageNet (Hall et al., 2020; Apostolopoulos et al., 2020).

Following Wang et al. (Wang et al., 2020b), we study classification for two CT datasets. To overcome the domain shift, their approach adds a contrastive loss, which reduces differences between latent space distributions. Unlike previous work in the domain of CT diagnosis of COVID-19, our method employs a generic ResNet architecture, and our contribution relates solely to the training loss and the inference-time augmentation procedure.

**Data augmentation** Many augmentation approaches were developed over the years as a form of regularization. These include geometric transformations (Taylor and Nitschke, 2017) and color space transformations (Wu et al., 2015), which have shown to improve many medical applications (Litjens et al., 2017).

Data mixing approaches create virtual samples that combine multiple images from different categories. The generated image has a fuzzy label from the two categories. In MixUp (Guo et al., 2019), the augmented image is a linear interpolation from two different images. The fuzzy labels are computed using the same weights as the images. Cutmix (Yun et al., 2019) extracts a box from one image and pastes it to the second. The fuzzy labels are proportional to the area of the box. SnapMix (Huang et al., 2021) is similar to Cutmix, except that the area of the patch is replaced by the sum of CAM activations within extracted and masked patches. It was shown to be highly effective on fine-grained classification datasets of natural images. Here, it is applied to the binary classification.

**Explainability** The task of generating a heatmap that indicates local relevancy from the perspective of a CNN observing an input image has been tackled from many different directions, including gradient-based methods (Shrikumar et al., 2017; Srinivas and Fleuret, 2019; Selvaraju et al., 2017), attribution methods (Bach et al., 2015; Montavon et al., 2017; Nam et al., 2019; Gur et al., 2020; Chefer et al., 2021), and image manipulation methods (Fong et al., 2019; Fong and Vedaldi, 2017; Lundberg and Lee, 2017).

The CAM method (Zhou et al., 2016) is based on the gradient of the loss with respect to the input of each layer. CAM and its extension, GradCAM (Selvaraju et al., 2017) have been used by downstream applications, such as weakly-supervised semantic segmentation (Li et al., 2018). Here, we use CAM in a novel way, to create more effective patch embeddings and drive the test time augmentation.

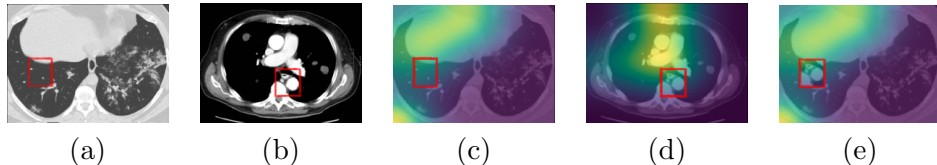

|     |     |     |     |     |
|-----|-----|-----|-----|-----|
| (a) | (b) | (c) | (d) | (e) |

Figure 1: An illustration of the SnapMix process. (a) A first random image, in this case a positive image from site-A, (b) a second random image, shown as a negative image from site-B. A random box is marked in each image. (c,d) the CAM maps of (a,b) respectively, with the associated boxes marked. (e) The SnapMix training image obtained by combining images (a) and (b) based on the random boxes. The label of the virtual training image is determined by mixing the labels of the two source images according to the sum of CAM activations in each box.

**Contrastive learning**    The loss we employ between patches of different levels of relevancy is related to contrastive learning methods, which have recently made a large impact in the field of self-supervised learning, where they are often used to link an image to its transformed version (He et al., 2020; Misra and van der Maaten, 2019; Chen et al., 2020). Our method is applied at the patch level. Contrastive learning has emerged in metric learning (Chopra et al., 2005) and subsequently in unsupervised representation learning (Hadsell et al., 2006). The learned embedding brings associated samples closer, while distancing other samples. In our case, association is determined by CAM-derived relevancy.

## 3. Method

Our experiments use a Resnet-50 network (He et al., 2016a) trained with the conventional binary cross-entropy loss $L_{\mathrm{BCE}}$ as baseline classifier. We then apply (i) SnapMix (Huang et al., 2021), (ii) a novel optimization term called Contrastive Patch Embedding loss, and (iii) a novel test time voting procedure. All three techniques use the heatmaps produced by the CAM method (Zhou et al., 2016).

### 3.1. SnapMix (Huang et al., 2018)

The SnapMix method is illustrated in Fig. 1. It combines two training images, depicted in panels (a) and (b), by considering a random box in each image (marked in red). The importance of each box is evaluated by integrating the CAM scores in them (panels c,d). The virtual sample is generated by pasting the box from the second image onto the selected box of the first image (panel e), and labeling the new image proportionally to the integrated CAM scores. More specifically, a ratio $(\rho_a, \rho_b)$ is computed for each image, by considering the sum of all CAM scores in a box over the sum of the CAM scores of the entire image. The labels are then linearly interpolated between the labels of the two images, using the the complement of the obtained box ratio in the first image $(1 - \rho_a)$ and the ratio in the second image $\rho_b$.

Unlike the original experiments of Huang et al. (2021), which considered datasets with many classes, in our case the problem is binary. It often happens that both images belong to the same class. Moreover, since we train using images from two sites, the virtual images created could play a role in overcoming the domain shift.

### 3.2. Contrastive Patch Embedding

The input images we receive are of size $224 \times 224$, the receptive field of the ResNet-50 architecture is of size 32, and the spatial dimensions of the embedding are $7 \times 7$, with a depth of $2,048$. For each of the $7 \times 7 = 49$ vectors in $\mathbb{R}^{2048}$ we compute the sum of CAM activations in the associated patch of size $32 \times 32$. We then select four vectors out of the 49: two with the highest sum of activations $u_1$ and $u_2$, and two with the lowest sum $v_1$ and $v_2$. The embedding loss we propose is a contrastive loss (Wu et al., 2018; He et al., 2020; Oord et al., 2018) that considers the dot products of the four vectors.

$$L_{\mathrm{CPE}}(u_1, u_2, v_1, v_2) = -\ln \frac{\exp(u_1^\top u_2)}{\exp(u_1^\top u_2) + \sum_{i,j=1}^{2} \exp(u_i^\top v_j)} - \ln \frac{\exp(v_1^\top v_2)}{\exp(v_1^\top v_2) + \sum_{i,j=1}^{2} \exp(u_i^\top v_j)} \tag{1}$$

This loss brings together the two most label-supporting embedding vectors and two most label-opposing embedding vectors. At the same time, it distances the top label-supporting embedding vectors from the pair of vectors that support the alternative label.

We note that the loss can be readily extended to any number of most label-supporting and most label-opposing vectors. This option is studied in our experiments, showing that there is no advantage in using more than two of each.

### 3.3. CAM-Directed Test Time Augmentation

It may be the case that the decision for a certain label is based on local artifacts that bias the network into making the wrong prediction. To avoid such cases, we classify each image $k + 1$ times: using the entire image, and masking one out of $k$ different patches.

For this purpose, we divide the image into small, non-overlapping patches of size $8 \times 8$, obtaining a grid of size $28 \times 28$. For each cell in the grid, we compute the sum of CAM activations. We then create $k = 31$ alternative images, by masking out sequentially the $k$ patches with the highest sum of activations. In the first alternative image, we mask out the patch with the highest CAM scores; in the second, we also mask out the second patch with the highest CAM scores; and so on. See Fig. 2 for an illustration.

The label we report is obtained through voting among the classifier output of the $k$ images. A supporting vote occurs when the pseudo-probability obtained from the network classifier is at least $\theta = 0.2$ if the original image has a positive label (i.e., a pseudo-probability larger than 0.5), or lower than $1 - \theta$ for images with negative labels. If more than half the $k$ votes are not supporting, we flip the label. In other words, if the inferred labels assigned by the classifier to the entire image are contradictory for more than half of the $k$ alternative images, with a high certainty, we flip the predicted label of the image.

## 4. Experiments

**Data** We evaluate the proposed method on three COVID CT datasets. For the first two, we follow the benchmark protocol and splits of Wang et al. (2020b). The SARS-CoV-2 dataset (denoted as site-A) consists of 2,482 CT images of 120 patients, of whom 1252 are positive with COVID-19. The 1,230 negative samples are affected by other lung diseases.

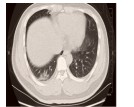 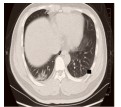 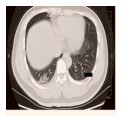 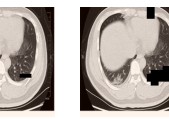 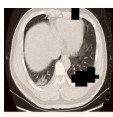 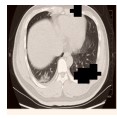 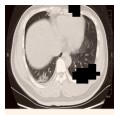

(0, 0.47)  (1, 0.61)  (2, 0.59)  (3, 0.64)  (28, 0.90)  (29, 0.90)  (30, 0.93)  (31, 0.92)

Figure 2: The Positive-label original test time image (left) obtained a negative classification score, with a probability of 0.47 of being positive. Removing even a small number of patches (Images 1-3) increased this probability to over 0.5. Subsequently, as more and more patches were removed, the probability of a positive case increased further, and became higher than $1 - \theta$; see the last derived images (out of $k = 31$ images), Images 28-31.

The resolution of these images varies between $119 \times 104$ and $416 \times 512$. The COVID-CT dataset (Zhao et al., 2020) (denoted as site-B) is much smaller, consisting of 349 CT images of 216 COVID-19 positive patients and 397 CT images of 171 control patients. The resolution of the images of site-B ranges from $102 \times 137$ to $1853 \times 1485$. Following (Wang et al., 2020b), the images of both datasets are resized to a fixed resolution of $224 \times 224$ and are intensity-normalized to zero mean and unit variance. Classification accuracy, F1 score, Sensitivity, and Precision are reported as percentages, using the train/test splits of the different datasets. The third dataset is COVIDx-CT (Gunraj et al., 2020), considered one of the largest in terms of the number of annotated samples provided. It contains 35996 training images of negative samples and 82286 of positive samples. The test split contains 12245 and 6018 samples for positive and negative patients, respectively. Following previous work on this dataset (Gunraj et al., 2020), we report accuracy as well as sensitivity and PPV (positive predictive value) for each infection type, at the single scan level. In all three datasets, multiple scans were obtained from the same patient, but treated as separated samples. Care was taken such that no patient would appear in both the train and test splits. See appendix A for more information.

**Implementation Details**     The architecture of our model[1] is based on ResNet50, followed by an MLP classifier. The ResNet model is initialized with pretrained ImageNet weights. We train the model for 200 epochs. The cross-entropy loss is used unweighted on the original samples or on virtual SnapMix samples, as dictated by a beta distribution with a parameter of $\alpha = 1$, which is the default parameter in (Huang et al., 2021). The $L_{\text{CPE}}$ loss is applied to all samples and is summed, unweighted with the cross-entropy loss.

**Baseline methods**     The first two baseline methods used for sites-A and B are methods that address domain shift in medical images. Series Adapter (Rebuffi et al., 2017) and Parallel Adapter (Rebuffi et al., 2018) include a domain adapter model based on a filter bank, in order to learn a joint representation from multiple datasets. MS-Net (Liu et al., 2020) was originally developed for a multi-site prostate segmentation task. It uses domain-specific auxiliary decoders. For classification tasks, each site is associated with an auxiliary classification head. The results of all three methods are from (Wang et al., 2020b).

The single and joint methods from (Wang et al., 2020a), employ an architecture called Covidnet. The difference lies in whether the method is trained on each dataset separately or not. It was also rerun in (Wang et al., 2020b), using a modified architecture (redesign).

---

1. Our code available at https://github.com/AmeenAli/Explainability_COVID19

Table 1: COVID-19 classification results for site-A

| Method | Accuracy | Precision | Recall | F1 |
|---|---|---|---|---|
| Series Adapter (Rebuffi et al., 2017) | 85.73±0.71 | 90.98±0.79 | 81.91±2.61 | 86.19±1.65 |
| Parallel Adapter (Rebuffi et al., 2018) | 82.13±1.91 | 83.51±1.87 | 80.02±2.47 | 82.39±1.78 |
| MS-Net (Liu et al., 2020) | 87.98±1.31 | 93.78±2.76 | 84.91±2.83 | 88.73±1.20 |
| Single (Covidnet) (Wang et al., 2020a) | 77.12±0.98 | 80.04±2.87 | 70.97±2.37 | 76.03±1.13 |
| Single (Redesign) (Wang et al., 2020b) | 89.09±1.08 | 94.58±2.07 | 83.78±0.62 | 88.97±0.91 |
| Joint (Covidnet) (Wang et al., 2020a) | 68.72±1.94 | 68.27±1.21 | 69.41±3.91 | 69.17±1.93 |
| Joint (Redesign) (Wang et al., 2020b) | 78.42±2.19 | 80.82±1.05 | 74.07±3.16 | 77.86±2.01 |
| SepNorm (Wang et al., 2020b) | 88.76±0.78 | 95.46±0.74 | 82.97±1.66 | 87.88±0.81 |
| SepNorm + Contrastive | 90.83±0.93 | 95.75±0.43 | 85.89±1.05 | 90.87±1.29 |
| Baseline architecture | 89.68±0.46 | 95.02±0.40 | 83.99±0.51 | 89.13±0.47 |
| Baseline + CPE loss (ablation) | 91.71±1.21 | 97.02±1.65 | 85.13±1.34 | 90.03±0.61 |
| SnapMix (Huang et al., 2021) | 92.38±0.32 | 98.33±1.81 | 86.42±1.50 | 91.92±0.37 |
| SnapMix + Contrastive | 91.99±0.13 | **99.02±0.52** | 84.22±1.22 | 91.03±0.83 |
| SnapMix + CPE (ablation) | 95.73±0.07 | 98.97±0.33 | 92.49±0.47 | 95.59±0.12 |
| Our full method | **95.90±0.24** | 98.64±0.12 | **92.93±0.40** | **95.87±0.25** |

The SepNorm method of (Wang et al., 2020b) uses features that are normalized for each site separately. It is further augmented with a contrastive loss that minimizes the domain shift ("SepNorm + Contrastive").

We present results for the ResNet-50 based architecture used by our method ("Baseline architecture"), and study the effect of our CPE loss (Eq. 1) on it ("Baseline+CPE loss"). Results are also presented for augmenting this architecture with the SnapMix method. As additional ablations, we present results for SnapMix combined with either the contrastive loss of (Wang et al., 2020b) ("SnapMix+Contrastive loss") or with our CPE loss ("SnapMix + CPE"). Finally, we present our full method, which includes SnapMix augmentation, the CPE loss, as well as CAM-driven test time augmentation and voting. For the COVIDx-CT dataset we compare our method with the reported baselines in (Gunraj et al., 2020). The COVIDNet-CT baseline (Gunraj et al., 2020) was pre-trained on ImageNet (Deng et al., 2009) and later fine-tuned on a COVIDx-CT (Gunraj et al., 2020) dataset, using stochastic gradient descent with momentum (Qian, 1999). We also compare our model with existing models for image recognition (ResNet50 , EfficeintNet-B0 , NASNet-A-Mobile (He et al., 2016b; Zoph et al., 2018; Tan and Le, 2019)) on the COVIDx-CT dataset.

## 5. Results

The results are reported in Tab. 1 for site-A, and Tab. 2 for site-B. Evidently, for both sites, the baseline architecture is already competitive with the best method from the literature, which is SepNorm with Contrastive loss. For site-A, the baseline is slightly inferior; for site-B it is considerably preferable.

Adding the CPE loss (Sec. 3.2) improves results for both sites. So does SnapMix augmentation, by a larger extent. The two contributions are complementary, and adding both CPE loss and SnapMix produces considerably better results than either on its own in site-A. In site-B, the combination of both produces a slightly higher F1 score than either contribution alone. However, SnapMix by itself is slightly better in terms of the three other scores. The ablation done using the contrastive loss of (Wang et al., 2020b) combined with

Table 2: COVID-19 classification results for site-B

| Method | Accuracy | Precision | Recall | F1 |
|---|---|---|---|---|
| Series Adapter (Rebuffi et al., 2017) | 70.01±3.82 | 63.04±4.87 | 74.91±1.89 | 67.08±3.09 |
| Parallel Adapter (Rebuffi et al., 2018) | 74.93±1.83 | 79.84±1.75 | 71.81±2.47 | 73.46±1.68 |
| MS-Net (Liu et al., 2020) | 76.23±1.81 | 79.29±1.48 | 74.07±1.29 | 76.54±1.73 |
| Single (Covidnet) (Wang et al., 2020a) | 63.12±2.09 | 64.03±3.91 | 57.73±2.94 | 61.09±1.28 |
| Single (Redesign) (Wang et al., 2020b) | 77.07±1.92 | 79.48±0.96 | 74.69±3.91 | 77.04±2.17 |
| Joint (Covidnet) (Wang et al., 2020a) | 63.27±2.82 | 64.27±3.81 | 54.19±4.17 | 59.78±3.12 |
| Joint (Redesign) (Wang et al., 2020b) | 69.67±0.92 | 64.98±3.17 | 66.94±5.86 | 66.89±4.91 |
| SepNorm (Wang et al., 2020b) | 76.89±0.65 | 80.74±2.98 | 70.34±3.76 | 75.02±1.14 |
| SepNorm + Contrastive | 78.69±1.54 | 78.02±1.34 | 79.71±1.42 | 78.83±1.43 |
| Baseline architecture | 85.23±0.41 | 86.54±0.84 | 83.58±0.81 | 84.51±0.62 |
| Baseline + CPE loss (ablation) | 85.96±1.22 | 87.03±1.22 | 84.71±1.02 | 85.22±0.79 |
| SnapMix (Huang et al., 2021) | 87.56±0.41 | **88.76±0.53** | 85.19±1.02 | 86.85±0.48 |
| SnapMix + Contrastive | 87.03±0.35 | 88.33±0.85 | 84.22±0.79 | 85.72±0.65 |
| SnapMix + CPE (ablation) | 87.02±0.49 | 88.32±0.69 | 84.69±1.02 | 86.95±0.76 |
| Our full method | **88.76±0.26** | 87.44±0.42 | **88.48±0.19** | **88.25±0.22** |

Table 3: Classification results on COVIDx-CT dataset

| Method | Acc | Sensitivity | | PPV | |
|---|---|---|---|---|---|
| | | Non-Covid-19 | Covid-19 | Non-Covid-19 | Covid-19 |
| ResNet-50 (He et al., 2016b) | 98.7% | 98.7% | 96.2% | 97.8% | 99.1% |
| NASNet-A-Mobile (Zoph et al., 2018) | 98.6% | 97.9% | 96.8% | 99.6% | 97.1% |
| EfficeintNet-B0 (Tan and Le, 2019) | 98.3% | 97.8% | 95.8% | 98.7% | 98.6% |
| COVIDNet-CT (Gunraj et al., 2020) | 99.1% | 99.0% | 97.3% | 98.4% | 99.7% |
| Basline architecture | 98.7% | 98.7% | 96.2% | 97.8% | 99.1% |
| Basline architecture + CPE loss | 98.9% | 98.8% | 97.1% | 98.0% | 99.2% |
| SnapMix (Huang et al., 2021) | 99.0% | 99.2% | 97.8% | 98.6% | 99.3% |
| SnapMix + Contrastive | 98.9% | 99.0% | 97.9% | 98.7% | 99.2% |
| SnapMix + CPE loss | 99.1% | 99.3% | 98.2% | 98.8% | 99.2% |
| Ours (Full) | **99.5%** | **99.7%** | **99.7%** | **99.8%** | **99.8%** |

the SnapMix technique hurts F1 performance relative to SnapMix in site-A, by increasing precision at the expense of recall. On site-B, it reduces all four scores.

Our complete method, which adds the test time augmentation of Sec. 3.3 on top of SnapMix and the CPE loss, obtains the best accuracy, recall, and F1 score among all methods. Its precision is slightly lower than the best ablation method. However, the gap in performance in F1 score (which combines both precision and recall) is substantial in comparison to the ablation method with the highest precision (site-A - 5%, site-B - 1.5%).

Tab. 3 depicts the results for the COVIDx-CT dataset. Evidently, our method achieves superior performance over all reported baselines. In this dataset, where performance is almost saturated, the improvement is smaller in absolute figures. However, our method is able to cut the error rate of the best method by at least half for all scores.

**Parameter Sensitivity** SnapMix employs the default augmentation parameters prescribed by (Huang et al., 2021). The CPE loss is defined without the temperature parameter commonly used in other contrastive learning methods and employs the minimal number of patches. It is, therefore, virtually parameter-free.

The parameter sensitivity of CAM-driven test-time voting is explored in Fig. 3, in which performance without this voting ("SnapMix+CPE") is depicted as a dashed horizontal line. When varying the number of augmented images $k$ (panel a), we observe that for any value of $k$, there is a performance boost for site-B, and this is maximized between $k = 30$ and

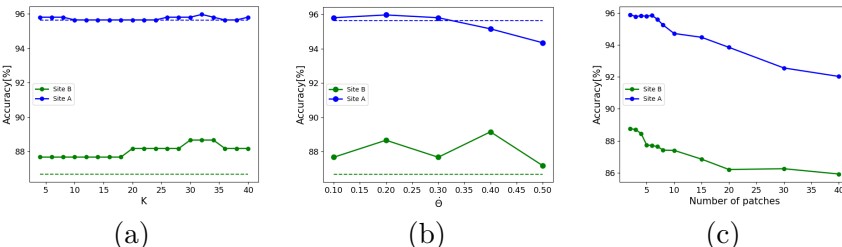

(a)                     (b)                     (c)

Figure 3: Ablation study showing (a) The effect of varying the number of alternative images $k$, (b) The effect of varying the certainty probability threshold $\theta$. (c) The effect of varying the number of patches in CPE loss. The dashed lines is panels (a) and (b) indicate the accuracy of a Resnet 50 with SnapMix + CPE without test-time augmentations.

$k = 35$. The performance boost for site $A$ is smaller for all $k$, and peaks at the value of $k = 31$. However, no value of $k$ hurts performance in site $A$.

Varying the value of the probability threshold, depicted in Fig. 3(b), shows that there is a positive benefit for all tested values $\theta \in \{0.1, 0.2, 0.3, 0.4, 0.5\}$ considering site-B. The largest contribution is for the value of 0.4. For site $A$, however, the contribution is positive only for conservative values (smaller than 0.3, when flipping the label of the test image becomes less frequent). The value of $\theta = 0.2$ provides a small boost to site-A and is also the 2nd highest for site-B.

Finally, the CPE loss we define employs, in each image, the two patches with the highest explainability score and the two with the lowest. It can be defined very similarly, to maximize similarity within groups of arbitrary size, while minimizing similarity between groups. This is explored in Fig. 3(c) for SITE-A and SITE-B. As can be seen, there is no advantage to using more than four patches in each image.

Appendix B presents qualitative visualization results on the behavior of the various classifiers as reflected by the explainability maps. Since the method is applicable to non medical datasets, Appendix C presents initial results for MNIST digit classification.

## 6. Conclusions

We present a method of COVID-19 detection in CT scans. The method tackles many of the challenges faced by medical imaging classification systems: distribution shifts across sites, limited training data, and the lack of region-based tagging. We propose combining three different techniques, which all rely on the heatmap produced by the CAM explainability method. The first method is a powerful regularizer called SnapMix, which was previously used for fine-grained classification. The second is a novel patch embedding method, which considers the two patches that show the strongest CAM activations in a given image and the two that present the lowest activations. Finally, we propose a voting method that constructs multiple masked images based on the CAM score. Taken together, our method obtains, despite using a generic network architecture, state-of-the-art results on publicly available COVID-19 CT datasets. The gap in performance is extremely large, and we demonstrate the individual contribution of each component to it.

## Acknowledgment

This project has received funding from the European Research Council (ERC) under the European Unions Horizon 2020 research and innovation programme (grant ERC CoG 725974).

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

## Appendix A. Datasets Splits

The dataset train/test splits that are employed are the exact splits specified by Wang et al. (2020b); Gunraj et al. (2020).

The **SARS-CoV-2 (Site-A)** dataset, provides with a total of 2482 CT scans, which are divided into 1252 positive samples and 1230 negative samples. In this dataset, the data is collected from 60 patients who are infected by covid-19 (32 males and 28 females) and the negative samples are based on 60 patients which are not infected with covid-19 (30 males and 30 females). The negative samples have other pulmonary issues. The splits used are 75%, 25% for training and testing respectively.

The **COVID-CT (Site-B)** dataset contains in total of 349 positives scans which are taken from 216 different patients, and 397 negative scans from 171 different patients without COVID-19. The splits used are 70% for training and 30% for testing.

The last dataset we employ is **COVIDx-CT**. It contains 35996 training images and 18263 testing images. These images were collected from 3745 patients.

**Preprocessing:** We do the following preprocessing for all of the datasets: all of the images are resized into 224x224 and later normalized to zero mean and unit variance.

**Validation set parameter sensitivity:** In the main text (Fig. 3) we consider the effect of varying the parameters of the augmentation loss and the parameter of the CPE loss on the test performance. We observe that the same parameters that are used in all of our experiments are effective for both sites.

Such a study is better done on a validation set. Since the datasets we employ do not contain a training set, we repeat the experiments when training on a random subset of 70% of the training samples, employing the rest of the training samples as the validation set.

The results are reported in Fig. 4. As can be seen, the results support the value for $\theta$ is 0.2, for $k$ is 31, and for the number of patches in the CPE loss is 3, for both of the datasets, Site-A and Site-B. Fig. 3 in the main text presents the analog results for the test set, which show a similar pattern

## Appendix B. Qualitative Explainability Results

Employing the explainability method of Gur et al. (2020), we present the heatmap explanations produced by applying SnapMix alone, SnapMix + our CPE loss, and the underlying architecture (resnet50) alone and with our CPE loss.

The first two rows in Fig.5, presents two cases in which the ResNet50 classifier failed to classify correctly, while the other methods managed to predict the correct label. On those two cases, we can see that the base architecture, focused on irrelevant areas differently from the other methods. The third and fourth rows are cases where all the methods predict correctly.

As can be seen for the negative cases, the SnapMix+CPE is distributed more uniformly over the lung regions, while for SnapMix and base+CPE, the map is focused on specific areas which is not the expected behavior for negative example. For the positive cases, we can see that the explainability map of SnapMix+CPE is sharper than the others, which can indicate more certainty with the decision.

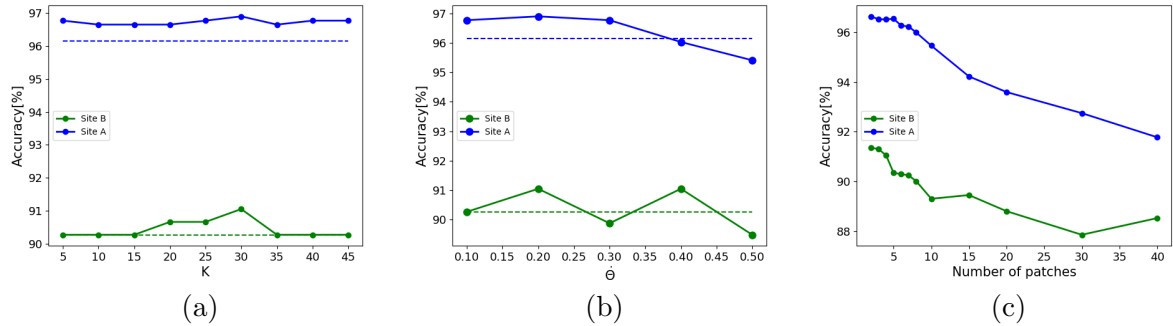

(a)            (b)            (c)

Figure 4: Hyper-parameters search - performances on the validation set, which is a training subset containing 30% from the images. (a) The effect of varying the number of alternative images $k$, (b) The effect of varying the certainty probability threshold $\theta$. (c) The effect of varying the number of patches in CPE loss. .The dashed lines is panels (a) and (b) indicate the accuracy of a Resnet-50 with SnapMix + CPE without test-time augmentations. The optimal value for $\theta$ is 0.2, for $k = 31$, and for the number of patches for the CPE loss is 2 - for both of the datasets, Site-A and Site-B.

Table 4: Result for domain adaptation for digit classification.

| Method | MNIST to USPS | MNIST to MNIST-M |
|---|---|---|
| CyCada (Hoffman et al., 2018) | 95.60% | - |
| LC + CycleGAN (Ye et al., 2020) | 97.10% | - |
| DRANet Bi-directional (Lee et al., 2021) | 98.20% | 98.70% |
| DRANet Tri-directional (Lee et al., 2021) | 97.60% | 98.30% |
| Resnet-50 (He et al., 2016a) | 98.13% | 98.70% |
| SnapMix (Huang et al., 2021) | 98.11% | 98.65% |
| Ours (Full) | **98.25%** | **98.92%** |

Figure 6 presents the shift of explainability maps during the test-time augmentation process. As can be seen, during the accumulated removal of the 31 patches, the explainability changes gradually to exclude these regions.

## Appendix C. Additional Results - Domain Adaptation

In order to check the suitability of our method beyond medical images, we consider the MNIST-to-USPS dataset. In which we train our model over MNIST dataset and test it over USPS dataset (Ganin et al., 2016) and the MNIST-M (Ganin and Lempitsky, 2015) dataset.

The USPS dataset is automatically generated from envelopes by the U.S. Postal Service containing a total of 9,298 16×16 pixel grayscale samples.

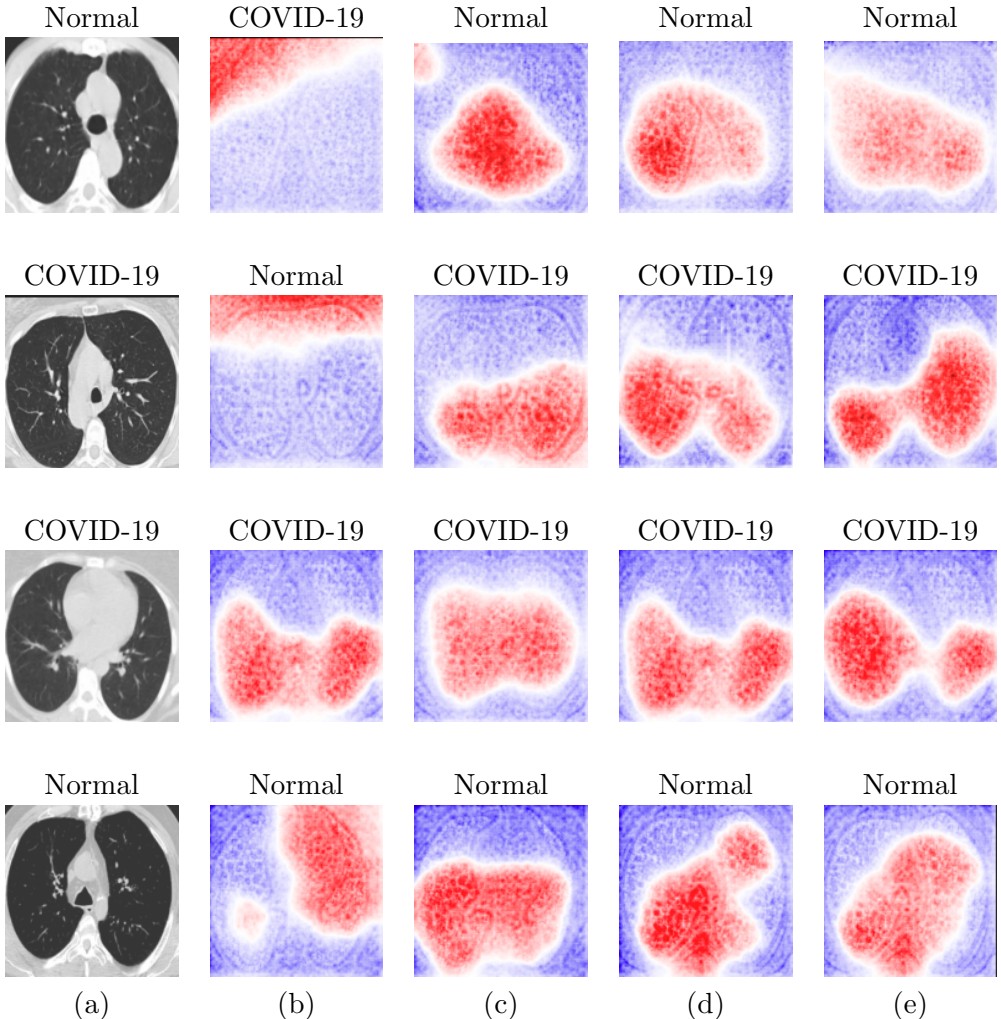

Figure 5: Visualization of explainability maps for four different methods where (a) The input CT-scan image (b) Resnet50 classifier - baseline architecture (c) Resnet50 classifier with CPE loss function (d) Resnet50 classifier with SnapMix augmentation (e) Resnet50 classifier with SnapMix augmentation and the CPE loss function. The correct label appears above the input image in column (a). The inference of each classifier is positioned above the classifier's heatmap.

MNIST-M (Ganin and Lempitsky, 2015) is created by merging the original MNIST digits dataset along with the patches which are randomly extracted from color photos of BSDS500 (Arbelaez et al., 2010) dataset as their background.

We compare our method with various recent baselines for Domain Adaptation, which include DRANet (Lee et al., 2021) that disentangles the feature representations into two elements, content and style. Domain Adaptation is then performed by applying the style features of the other domain. The method of Ye et al. (2020) calibrates the target domain images to better fit the source classifier's representation while maintaining the source domain

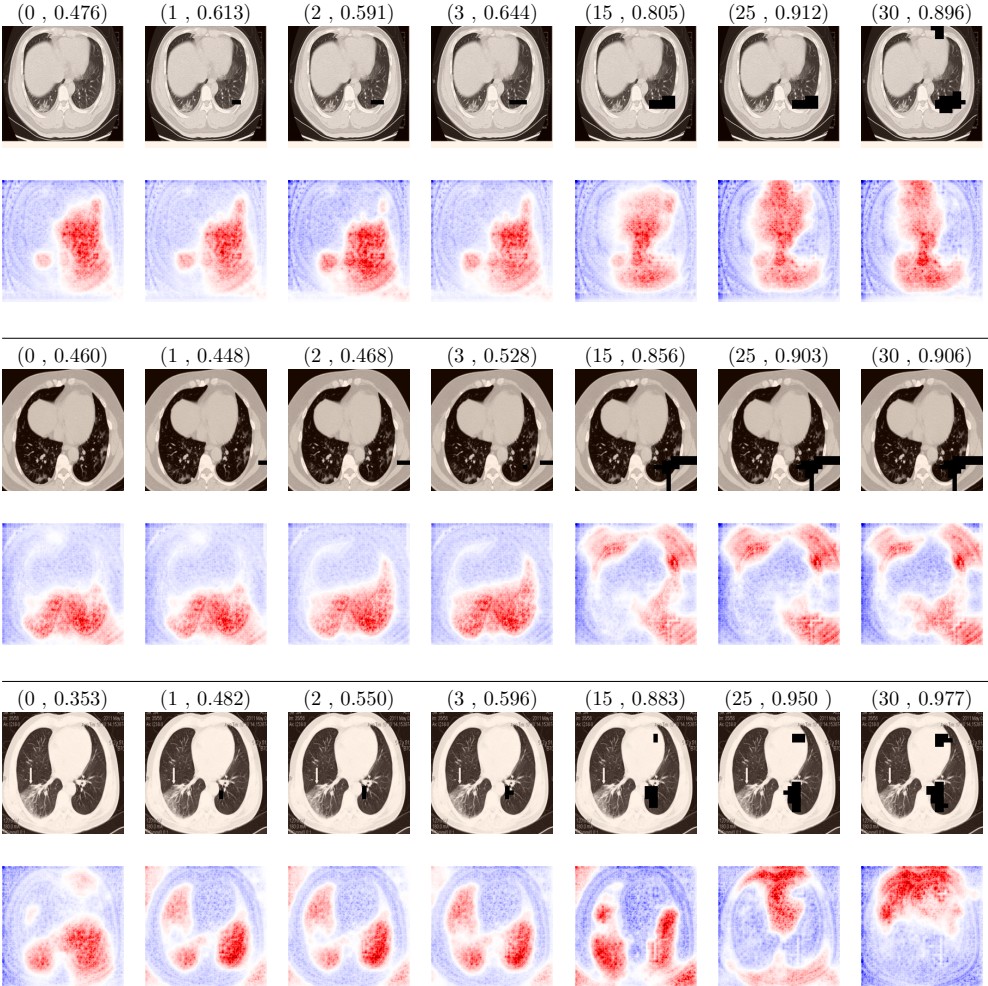

Figure 6: The effect of removing high relevancy patches during test-time on the explainability heatmap. In each segment, each column has three elements: a pair of numbers indicating the number of removed patches and the classification score, the augmented image, and the explainability map of the modified image. As can be seen, removing high relevancy patches alters the explainability map in a smooth (frames are skipped at the high values of $k$) yet significant way.

performance. Hoffman et al. (2018) employ a cycle-consistent adversarial domain adaptation method.

As ablations to our full method, we also compare with ResNet50 and ResNet50+SnapMix.

As can be seen from the results in Table. 4, our method is outperform compared to the recent baselines in the domain adaptation settings.

