# OpenReview forum: "Explainability Guided COVID-19 Detection in CT Scans"
_MIDL.io/2022/Conference — MIDL 2022_

### Official Review · Reviewer_4J9Y · 2022-01-05

**Confidence:** 4
**Preliminary Rating:** 4
**Recommendation:** Poster

**Summary:**

This paper adapts the SnapMix data augmentation method to improve classification of COVID-19 CT image slices. The method appears highly successful at improving over the presented baselines using several open datasets. Methods are presented in sufficient detail, but the dataset and validation approach are less clearly articulated.

**Strengths:**

* The primary contribution of this method is application of the SnapMix data augmentation approach with a contrastive loss function and a voting fusion procedure.
* Extensive baseline procedures are examined and substantial improvements are seen.
* The method is highly competitive with the state of the art.

**Weaknesses:**

* There is limited methodological novelty.
* A statistical assessment of paired differences was not performed.
* No discussion of the 3-D nature of CT images was present in the data section.
* For methodological development, the paper falls short as it focuses on a single task. Hence, I have categorized this paper under validation / application.

**Deanonymize Review:**

no

**Final Rating After The Rebuttal:**

4: Weak Accept

**Justification Of The Final Rating:**

The additional experiments and the appendices improve the quality of the manuscript and reaffirm the moderately positive assessment of the initial version. The reliance on appendices for key technical details is less helpful.

**Paper Type:**

validation/application paper

**Questions To Address In The Rebuttal:**

It is important to document that the test/train splits were performed on a patient basis and a single patient does not exist on both sets. It is unclear if the tables are reported on a patient basis or on a slice basis. Clarity on the sampling with respect to the patient distribution and slices per patient are important.

**Special Issue:**

no

---

### Official Review · Reviewer_1NXN · 2022-01-21

**Confidence:** 3
**Preliminary Rating:** 4
**Recommendation:** Oral

**Summary:**

Authors present their method for binary image classification, in which they evaluate the addition of three 'explainability-based' components: data augmentation via SnapMix, a loss function based on contrastive patch embedding, and test time augmentation. All three components are based on the CAM method (class activation mapping). The results show that adding the three components does improve classification of covid in CT images, when compared to both results reported in literature and baseline implementations.

**Strengths:**

The paper is clear and, personally, I think the CAM based additions to the image classifier are of interest. Authors included a clear description of the methods; and exhaustive results from both literature and their own baseline experiments. They included quite a large amount of data for their experiments.

**Weaknesses:**

Personally, I think COVID classification on medical images is not a useful task. Detection of COVID is easily done with nasal swabs (pcr or self-test), which is also much cheaper than acquiring medical images. There is no added value in COVID detection on imaging. I think authors should include a more clear rationale for doing this specific task: do they actually want to improve covid detection or is this 'just' a dataset to demonstrate their method on?

Another weakness is the use of 2D image slices, instead of using actual 3D volumes. I do not understand why authors do not use 3D imaging, since they use CT images? CT imaging is by nature 3D, but authors reduce this to a 2D image classification task. However, it is not clear why and how the 2D slices were selected from the 3D volumes? Why is the method not applied in 3D? How are the multiple 2D classifications turned into a final classification on the patient level?

It is also not clear how the dataset was split into training / validation / test sets. Especially information about the validation set is missing. It seems that some of the experiments and parameter-tuning (e.g. in Figure 3) are performed on the test set, in stead of the validation set. This should be clarified in the manuscript.

**Deanonymize Review:**

no

**Detailed Comments:**

I think the title of the manuscript should be more descriptive.

I would suggest to include exact numbers / results in the Abstract. Something like 'a gap twice as large' is not very exact. Also, I think a little bit more information on the methods can be included in the Abstract.

In the last paragraph of the Introduction, I would suggest to include the exact numbers. Not vague quantifications like 'in the high 70's percentage' or 'reduce ... by one and a half or more'.

In the Experiments, please include a clear description on the 2D / 3D nature of the data and how slice-selection choices were made.

Please make a separate Results section, in stead of including the results in the Experiments section. Also consider a separate Discussion section, so that the Results section remains clear and factual.

**Final Rating After The Rebuttal:**

4: Weak Accept

**Justification Of The Final Rating:**

I would like to thank the authors for providing their comments and clarifications. Based on that, I have decided to keep the original rating I had provided.                                              q

**Paper Type:**

validation/application paper

**Questions To Address In The Rebuttal:**

What is the rationale for COVID detection on medical images?
How will this impact clinical routine?
Why are 2D slices used (if so?) and not 3D images?
Will the methods work in 3D?
Can other explainable methods that generate saliency maps be used in stead of CAM?

**Special Issue:**

no

---

### Official Review · Reviewer_QW8f · 2022-01-24

**Confidence:** 3
**Preliminary Rating:** 4
**Recommendation:** Poster

**Summary:**

In this paper, the authors tackle three problems in medical image classification: limited dataset, lack of region-based supervision and domain shift. For this purpose, the data is augmented using previously proposed approach Snap-Mix, a new patch embedding technique is employed that considers the two patches each with the highest and lowest CAM activations and a voting method is investigated that constructs multiple masked images based on the CAM score. The approach is evaluated for the classification of COVID-19 in chest CT images using three datasets.

**Strengths:**

- The authors provide through validation of the proposed method using three large covid-19 chest CT datasets. The method is compared with previously proposed approaches and the parameter sensitivity is investigated in details. The results show that the proposed approach results in improved performance for all three datasets.
- The paper is easy to follow and the ideas are supported by visualizations.
- Reproducibility aspect: Implementation is made available in public.

**Weaknesses:**

- Paper lacks clarity on several important aspects: distribution of dataset into train/test/validation, how information from 2D slices is merged to finally obtain patient-level image classification etc.
- The formatting of the paper needs some improvement (For example, repositioning figures and tables, avoiding casual writing etc. For more explanation, please see in the section below).

**Deanonymize Review:**

no

**Detailed Comments:**

- How does theta value of 0.2 in the test time augmentation relate to a pseudo-probability of 0.5?
- For the CAM based test time augmentation, the highest CAM activated regions are masked out considering that the activations come from artefacts. But otherwise, the most significant part of the image where the network decision for the classification comes from is being ignored. Probably the motivation needs some explanation.
- Section 3.3 / Conflicting information: First it is mentioned that image is classified k+1 times by masking one out of k different patches. Later, it is explained that "in the second, we mask out the two patches with the highest CAM scores; and so on".
- Avoid casual sentences: "in which performance levels are in the high 70’s percentage", "we reduce the error rate by one and a half or more, depending on the measurement error" (what is meant by "depending on the measurement error"?)

Minor comments:
- Maintain uniformity: state-of-the-art vs state of the art, Site-A vs SITE-A, Snap-Mix vs Snap-Mix
- Change "which have been shown" to which have shown
- Not clear, maybe needs rephrasing: "The first two datasets are trained together on the literature"
- Dataset A: It is mentioned that "The 1,230 negative samples are affected by other lung diseases". Are all 1,230 negative samples affected by other lung diseases? Otherwise, the sentence needs to be rephrased.

**Final Rating After The Rebuttal:**

4: Weak Accept

**Justification Of The Final Rating:**

The authors addressed several important questions in the rebuttal phase and added relevant details/illustrations to improve the paper. In my opinion, the paper could be useful to the community for exploring class activation mappings for improved image classification. For the practical application in clinical practice, the authors could investigate more on how to derive patient-level decision with the proposed technique with minimal overhead on the inference time.

**Paper Type:**

validation/application paper

**Questions To Address In The Rebuttal:**

- Reposition figures 1 and 2 closer to the associated text to improve the readability. Similarly, reposition the tables 1-3 so that they are close to the referenced text.
- Figure 2: Additional explanation needed. What is the GT label? Reference to theta comes later in the text, probably need to adapt the text or the position of the figure.
- The statement is not clear and needs more explanation: "Unlike previous work in the domain of CT diagnosis of COVID-19, our method employs a generic ResNet architecture, and our contribution relates solely to the training and inference procedures"
- If possible, include qualitative evaluation for the final image-level classification or explanation of where the image classification goes wrong (if there are any trends).


**Special Issue:**

no

---

### Meta-Review · Area_Chair_d46H · 2022-02-14

**Recommendation:** Accept (Poster)
**Confidence:** 4

**Metareview:**

Three reviewers have reviewed the work. While not all reviewers are convinced about the clinical value of COVID detection in CT scans, they recognize that the work has methodological contributions, and the strength of the quantitative results is clear. The use of 2D slices instead of 3D CT volumes is mentioned as a limitation. The work is recommended for presentation as a poster.

---

### Decision · Program_Chairs · 2022-02-28

Accept